# Investigating Perceptions of Land Issues in a Threatened Landscape in Northern Cambodia

**Emilie Beauchamp [1,2,*], Tom Clements [3] and E. J. Milner-Gulland [2]** 

[1]  International Institute for Environment and Development, London WC1X 8NH, UK
[2]  Department of Zoology, University of Oxford, Oxford OX1 3SZ, UK; ej.milner-gulland@zoo.ox.ac.uk
[3]  Wildlife Conservation Society, Cambridge CB2 3QZ, UK; tclements@wcs.org
*  Correspondence: emilie.beauchamp@iied.org

**Abstract:** Land governance highly affects rural communities' well-being in landscapes where land and its access are contested. This includes sites with high land pressures from development, but also from conservation interventions. In fact, local people's motivations for sustainably managing their resources is highly tied to their perceptions of security, trust and participation in land management regimes. Understanding these perceptions is essential to ensure the internal legitimacy and sustainability of conservation interventions, especially in areas where development changes are fast paced. This paper presents an analysis of household perceptions of land issues in 20 villages across different conservation and development contexts in Northern Cambodia. We assess whether conservation and development interventions, as economic land concessions, influence perceptions of land issues in control and treatment sites by modelling five key perception indicators. We find that household characteristics rather than village contexts are the main factors influencing the perceptions of land issues. Interventions also affect perceptions, especially with regards to the negative effect of development pressures and population growth. While large-scale protected areas do not calm insecurity about land issues, some village-based payment for environmental services projects do. Ultimately, evidence from perception studies can help address current concerns and shape future conservation activities sustainably.

**Keywords:** perception indicators; land issues; payment for environmental services (PES); economic land concessions; protected areas; community well-being; Cambodia

## 1. Introduction

Issues related to management, security and access to land highly affect the rural communities' well-being in developing countries, especially in contested conservation landscapes where land and its access is often restricted [1–3]. In such context, local people's motivations for sustainably managing their resources and achieving conservation targets is highly tied to their perceptions of well-being [4,5]. This is because locally valued resources such as land have material but also relational and symbolic dimensions [6].

In this context, subjective evaluations, or perceptions, of the multiple dimensions pertaining local community well-being are important to understand, in order to appreciate the potential trade-offs experienced within and between communities because of conservation [4,7]. This is particularly the case in landscapes where conservation spatially overlaps with other development programmes and external pressures that can negate or even erode the intended effects of conservation on local communities' environments, institutions and well-being [8–10].

While an abundance of literature has aimed at assessing the social outcomes of conservation over the past decades [11], research has primarily focused on the material, objective dimensions of

human well-being [12,13]. To date qualitative case studies have dominated the evidence based on relational and subjective aspects of human well-being in a conservation context [14]. Inferences from these studies may have power to explain and contextualize effects in complex and overlapping land systems, yet non-statistical attributions from a small number of sites can be unreliable and lack the transferability needed to support policy decisions [12,15]. Outcomes from the subjective dimensions of human well-being, based upon assessing individual perceptions, have rarely been reliably documented quantitatively at a landscape scale [16,17]. Locally defined perceptions indicators can thus provide important insights into the issues underlying the social impacts of conservation, such as the internal legitimacy and the acceptability of conservation rules and institutions, and how these interact with other interventions [18,19].

This paper presents an analysis of factors influencing the household perceptions of land issues in 20 villages located across different conservation and development contexts in Northern Cambodia. We assess whether the presence of protected areas (PAs) and economic land concessions (ELCs) influences the local perceptions of issues related to land use. While the overlap between development and conservation interventions can be observed in several developing countries, Cambodia presents an interesting case study for the analysis of these questions considering the increased competition for land resources it had experienced over the past decade [20].

We focus on perceived current and future access to land, participation in land management decisions and trust in local authorities that implement land rules, as they represent important aspects of locally defined human well-being conceptualisations in communities in Northern Cambodia [21,22]. We ask: what are the factors influencing perceptions of well-being related to land issues across different settings of competing land use? Second, how does the presence of conservation and development interventions, in terms of PAs and ELCs, affect these perceptions? Third, how do perceptions vary for different groups between and within villages? Last, how do these perceptions vary between the wider landscape and villages where community-based payments for environmental services (PES) programmes have been implemented?

Availability of productive land is a central necessity for the livelihoods of most families in Northern Cambodia. Given the recent rush to acquire land resources and the consequent decrease in land availability for rural families, because of population growth and competition with ELCs [23], we expect perceptions of land issues to vary between PAs, where land expansion is controlled, and outside PAs. Given the negative environmental and social impacts of ELCs reported, we expect ELC presence to strongly undermine the perceptions of access to land [20,24,25]. However, we expect perceptions of trust in land authorities to be higher inside PAs, where local institutions have been supported through the PA intervention over the past nine years.

*1.1. Study Site*

Cambodia has seen rapid development and globalisation over the past decade [26,27], paralleled by a similarly high rate of resource depletion in terms of deforestation rates and illegal logging [28–30]. The 2001 Land Law initiated a process of codifying land claims by local people, indigenous communities and the private sector, marking a move towards a formal land registration system and official land titles [31]. Thus while customary land rights, or 'possession rights', were recognized under the 1992 Land Law, they were replaced in the 2001 Land Law by modern landownership where titles are required [32]. This introduction can be beneficial in supporting tenure security for local communities [33], yet its weak enforcement and co-option by the elite opened the doors for the monetisation of land titles. It also enabled the widespread granting of economic land concessions (ELCs) as large areas of state public land were reclassified as state private, consequently weakening informal possession rights in poor rural communities [34].

The government of Cambodia's promotion of ELCs as a mechanism for agricultural intensification has raised increasing concerns about the impacts of their widespread granting and the lack of adherence to the established legal criteria in the granting process [24,34,35]. Issues have arisen specifically around

ELCs being a façade for land grabbing and causing unfair eviction of communities from their land and human rights abuses [35], with the number of land disputes increasing since 2001 [32].

The landscape of the Northern Plains of Cambodia represents an appropriate site to study the perceptions of land issues in a context of high and contrasting resource pressures, as it contains a combination of different levels of conservation interventions—protected areas (PAs) and payments for environmental services (PES)—as well as overlapping ELCS. The Northern Plains is a landscape located in the province of Preah Vihear along the border with Thailand and Lao (see Figure 1).

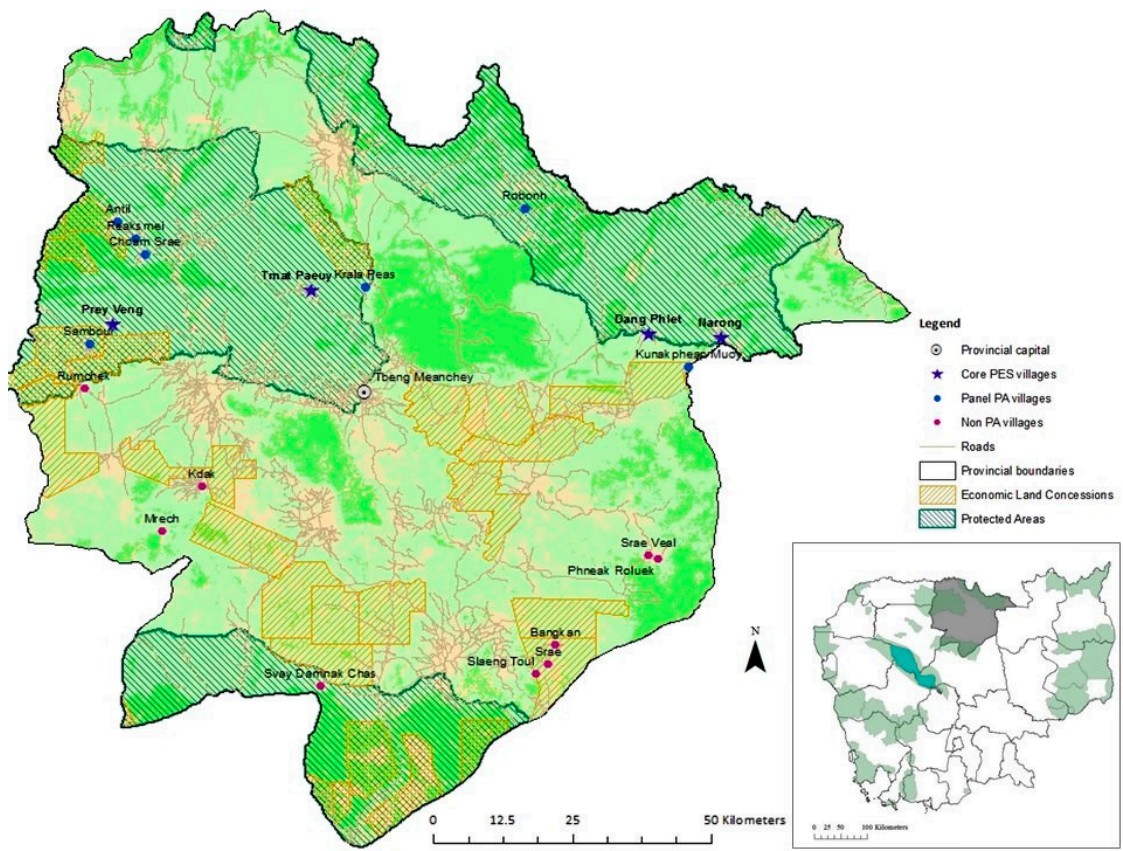

**Figure 1.** Locations of the surveyed villages, boundaries of economics land concessions (ELCs) and protected areas (PAs) in the province of Preah Vihear.

The study site is one of the largest remaining areas of deciduous dipterocarp forest and is considered an area of high biodiversity interest [36,37]. Rural households in the study site are subsistence farmers whose livelihoods revolve around small-scale rice farming, with additional non-timber forest product harvesting and fishing around villages. Collecting liquid resin from Dipterocarp trees has also traditionally been an important livelihood in the Northern Plains' communities [38,39].

*1.2. Interventions in Study Site*

The landscape contains two managed PAs: The Kulen Promtep Wildlife Sanctuary managed by the Ministry of Environment and the Preah Vihear Protected Forest managed by the Forestry Administration of the Ministry of Agriculture, Forestry and Fisheries. The Preah Vihear Protected Forest was declared in 2002, and in 2018, after data collection, was split under two PAs: The Preah Roka Wildlife Sanctuary and Chhep Wildlife Sanctuary. The Kulen Promtep Wildlife Sanctuary was established in 1993 as part of Cambodia's first protected area network. Since 2005, international non-governmental organization Wildlife Conservation Society has assisted governmental agencies in their conservation efforts in both PAs [39]. One such ongoing effort since 2005 is community development assistance to develop participatory land use plans for PA villages, in order to gain official

status and formalize the customary tenure rights in place [39]. A third PA is present in the south of the landscape, however is unmanaged and considered largely as a paper park [40,41] The Boeng Per Wildlife Sanctuary (BPWS) is by law under the management of the MoE, yet has seen insufficient law enforcement activities and was identified as the protected area second most threatened by land encroachment in Cambodia [42].

Resource use rules within PAs under Cambodian law allows local uses such as non-timber forest product collection, although forest clearance, commercial logging, and hunting or trade in threatened species are illegal. The imposition of conservation resource use rules may prevent local communities accessing certain resources and from diversifying livelihoods, risking creating a dependency on subsistence use of forest resources and leading into a 'poverty trap' [43,44]. Within both PAs, households can expand agriculture within the agreed boundaries of a locally developed participatory land use plan. The creation of new settlements within PAs is forbidden, and the number of households allowed to migrate to PA villages is limited.

Additionally, the Wildlife Conservation Society has supported the government of Cambodia's agencies in implementing three types of PES schemes within the two PAs: a bird nest protection programme, a premium payment scheme for eco-friendly rice (Ibis Rice) and an ecotourism programme [40,42,45]. The PES schemes were designed in response to a high level of threat where conservation opportunity costs, at least for conversion of forest lands, were also moderately high [41]. Implementing PES in the context of weak institutions can be difficult [46], and projects worked towards strengthening local village institutions and land rights, for example through the development of the participatory land use plans [42,45]. In each PA village, a locally elected community protected area (CPA) committee manages the compliance to rules for participating households and oversees that PA rules are respected around the village. This committee is elected through village elections every five years and oversees payments as part of the schemes along with compliance to the PES and PA rules with the support of the Wildlife Conservation Society.

The study landscape also counts 30 ELCS accounting for a total of 242,505 hectares; 77,175 of which are located over designated PA areas [47]. Because of the abolishment of customary rights in 2001, the informal tenure rights of Cambodian rural families are legally superseded by the ownership rights granted to ELCs by the government of Cambodia [20,48]. Therefore, the impacts of ELCs are often felt disproportionately in rural areas, where land and resources technically still owned and managed by the state yet where land laws are indirectly implemented by local commune councils, thus making villagers' rights unclear [41]. ELCs have also been reported to be granted over high value forests and over protected areas, driving Cambodia's high deforestation rates [24,49,50]. The contribution of ELCs to local rural economies comes further into question with studies showing that most ELC workers are migrants from outside the province [51].

## 2. Materials and Methods

### 2.1. Survey Design

Quantitative data were collected as part of a wider impact evaluation which surveyed 1129 households across 20 villages in the province of Preah Vihear between July and December 2014 [52]. The survey design for data collection builds on a historical dataset of 11 PA villages and 9 counterfactual villages selected through co-variate matching using four key factors characterizing village-level poverty prior to the initiation of the PAs at a 2005 baseline [40]. Five counterfactual villages located at least 20 km away from the PA border were retained for the purpose of a multi-period impact evaluation of objective well-being indicators [53]. The 2014 data collection added four new villages that were among the matched results from the original quasi-experimental design, but that had not been surveyed in 2008 and 2011. These villages, Rumchek, Bangkan, Srae and Slaeng Toul, were added in 2014 because of the presence of ELCs close to their village, thus providing an interesting contrast against which to assess the influence of conservation.

In a dynamic environment it is rare that all control and treatment units evolve with similar trends apart from the intervention over a medium-term period, as is required for statistical inference to be made based solely on matching without longitudinal data. This is especially the case in the study landscape where a large number of ELCs have been granted since 2008. Out of the 11 villages located in PAs, five are now affected by ELCs; while six of the nine villages located outside PAs are affected by ELCs (see Figure 1 and Appendix A for village information). Therefore, this study represents a landscape-scale comparison, rather than a formally matched design. While the three PES projects have been implemented at different intensities across the 11 PA villages, the four villages that have been the focus of higher conservation activities by WCS since 2008 are used to assess the additionality of effects of PES in PA areas [21].

The questionnaires contained sections on households' demographic information, their livelihood strategies and economic status, and their perceptions of land issues. Perceptions and subjective well-being issues are often at risk of being misunderstood [52], thus translation of the concepts was done by the research team along with translation professionals. All research protocols were approved by Imperial College Research Ethics Committee before the start of the research.

*2.2. Variables and Statistical Modelling*

The perception indicators used in this study were developed following extensive work in the communities defining individual conceptualisations of well-being [21]. While the multidimensionality and geographical heterogeneity of well-being conceptualisations were apparent, agricultural land featured as the most prominent well-being item across all villages and population sub-groups (see Figure 2). Based on this work, five perception indicators relating to trust, management, security and access to land were developed (see Table 1). The questions were phrased as an opinion statement with a choice of six answers along a Likert scale going from: (5) 'strongly disagree', (4) 'disagree', (3) 'neutral', (2) 'agree', (1) 'strongly agree', adding (6) 'I do not know/not applicable/do not wish to answer'. The latter were excluded from the regression analysis. Each indicator thus represents the respondents' perceptions as a response variable [54].

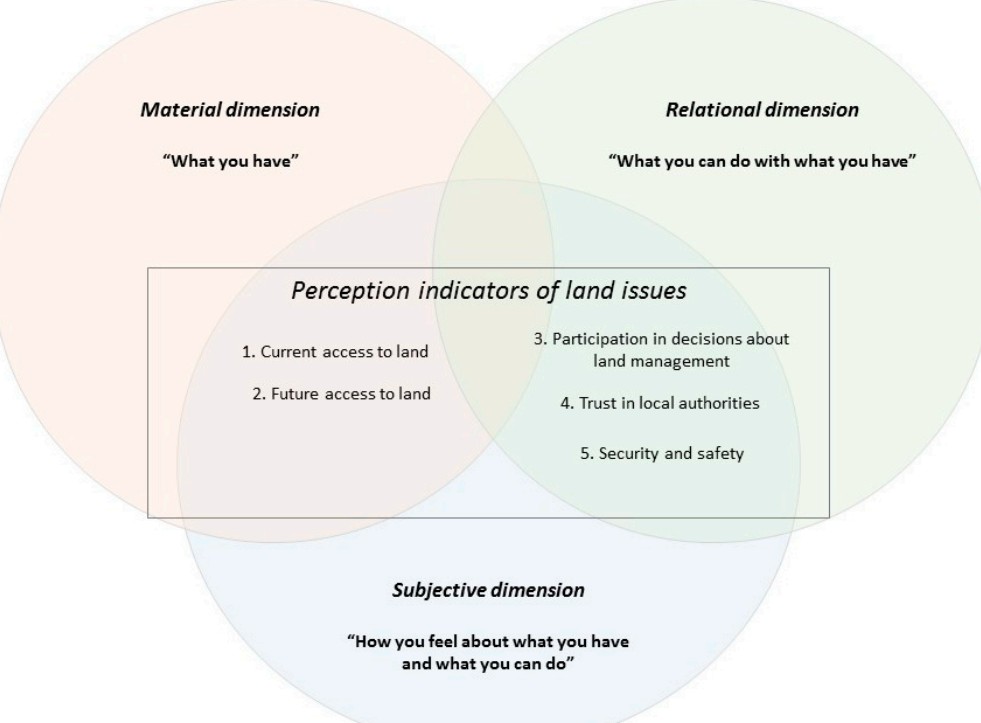

**Figure 2.** Perception indicators representing subjective evaluations of well-being issues related to land.

**Table 1.** Examples of opinion statements for eliciting household perceptions on issues of access to, participation and trust in management of land.

| Perception Indicator | Example Statements |
|---|---|
| Access to resources | 'You feel that you will have enough access to land for your household needs in three years' time' <br> 'You feel that you currently have enough access to land for your household needs' |
| Participation in management | 'You feel that you are involved in decision-making about land use management in your village' |
| Trust in authorities | 'You feel that you can trust the commune council to implement land laws and policies in your village' (all villages) <br> 'You feel that you can trust the CPA committee to protect the forest and implement land laws in your village' (PES villages) |

Likert-type indicators were modelled as response variables to the binary treatments of PA and ELC presence, controlling for other a priori external variables (Table 2). While PA presence can be straightforwardly determined by the village location inside or outside the PA, not all declared ELCs are active, or even active across their entire area [55]. Presence of ELCs was thus determined as whether an active ELC was located within 3 km of the village, with confirmation from field surveys and reports from villagers. In this sense, ELC presence was determined by geographical presence along with villagers' perceptions of ELC presence and based on previous research (see Appendix A) [47].

Perceptions are also linked to demographic and socioeconomic attributes such as gender, age, economic status and level of resource dependence of the respondent's livelihood [18,56]. Therefore other a priori predictor variables included are factors hypothesised to influence perceptions of trust, management, security and access to land between and within villages in 2014. Economic status was calculated using the basic necessities survey (methodology, which incorporates multiple aspects of poverty into a single score for each household in the sample) [57,58]. Land issues often reflect landscape dynamics and we have also included a village-level variable to reflect the level of village development through the proxy of the number of years of schooling available in the village. Access to formal education can be a useful proxy for the level of development, with schools being an epicentre for human capital and nexus of population movement [59–61]. In fact, villages with a higher level of education available to residents, such as Bankan, Dongplat and Tmatboey, are also more populated compared to the neighbouring villages and represent a hub in their respective commune for commerce and transport connections (see Appendix A for village information).

Presence of PAs and ELCs as treatment variables were replaced with binary variables indicating for participation in each of the three PES projects between 2008 and 2014 for the subset of models addressing the effect of PES participation on perceptions of land issues, for the four villages with the most intensive PES interventions. A compliance variable was added to the PES models, representing the number of years between 2008 and 2014 a household had broken PA rules and illegally cleared new land. The expectation was that those households which had not complied with the rules may have lower levels of trust in authorities, perceived access to land and involvement in decisions.

*2.3. Statistical Modelling*

Collinearity was tested using a correlation matrix and a variable inflation factor test and continuous variables were square-rooted to improve model convergence. We use ordered logistic mixed regression models to analyse the Likert scale indicators as response variables. We adopted a four-step procedure to generate a final model of probability for levels of response. First, we fitted a full (global) model with the a priori fixed effect variables and proceeded to variable selection using backward selection from full models including all supported variables, using AICc values and removing variables that did not improve the model, as represented in an AICc > 4. Once the most parsimonious fixed effect model was

selected, we evaluated the influence of intra-village correlation by comparing the fit of adding 'village' as a random effect against the fixed effect model only, using a likelihood-ratio $\chi^2$ test. If significant, the final model were fitted with village as a random effect, which is assumed to act additively to the baseline of the log-odds function [62,63].

Third, final model validation was performed by confirming the accuracy of the error in the reported value of the log-likelihood through a convergence test, and by testing the proportional odds assumption using likelihood ratio test of equal slopes [64]. Last, the convergence properties of the fitted model were illustrated by plotting slices of the log-likelihood function; and the relative profile likelihood of the parameters within confidence intervals were also plotted to check the symmetry of the confidence intervals and illustrate effects of parameters in the fitted model. If the final model was multilevel, the conditional modes of the random effect were plotted with 95% confidence intervals based on the conditional variance. The package 'ordinal' was used to fit the model [64].

**Table 2.** Predictor variables to be included in full models.

| Variables | Description and Comments | Type | Full Dataset | PES Subset |
|---|---|---|---|---|
| *Interventions* | | | | |
| Intervention PA | Presence of PA | Binary: 0: No; 1: PA | √ | |
| Intervention ELCs | Presence of an active ELC within 3 km of village; confirmed ELC disturbances | Binary: 0: No; 1: ELC | √ | |
| *Individual characteristics* | | | | |
| Head of household age | | Cont. (sqrt+1) | √ | √ |
| Head of household gender | | Binary—M/F | √ | √ |
| Head of household education | | Cont. (sqrt+1) | √ | √ |
| *Household characteristics* | | | | |
| Household size | Total number of members | Cont. (sqrt+1) | √ | √ |
| Poverty (BSN score) | BSN score (2014 weights applied retrospectively) | Cont. (sqrt+1) | √ | √ |
| Land owned (ha) | Total land owned—paddy, chamkar and cash crop (hectares) | Cont. (sqrt+1) | √ | √ |
| *Resource use and livelihood strategies* | | | | |
| Resin-tapper | | Binary—Yes/No | √ | √ |
| Rice farmer type | 0: Both; 1: Paddy only; 2: Shifting cultivation only (chamkar); 3: None | Categorical | √ | √ |
| Cash crop farmer | | Binary—Yes/No | √ | √ |
| Employment | Employed by either private or public sector (army service excluded) | Binary—Yes/No | √ | √ |
| NGO employed | Employed by NGO | Binary—Yes/No | √ | √ |
| Service or shop provider | Service provider (rice threshing and milling excluded) or shop keeper | Binary—Yes/No | √ | √ |
| Sell labour | | Binary—Yes/No | √ | √ |
| *Village level variables* | | | | |
| Education in the village | Top year of school available in village | Cont. (sqrt+1) | √ | √ |
| Time to provincial capital | Dry season travel time to Tbeng Meanchey (h) | Cont. (sqrt+1) | √ | √ |
| *PES subset level variables* | | | | |
| Ecotourism participant | Received Ecotourism payment between 2008–2014 | Binary—Yes/No | | √ |
| Ibis Rice participant | Received Ibis Rice payment between 2008–2014 | Binary—Yes/No | | √ |
| Bird Nest participant | Received Bird Nest payment between 2008–2014 | Binary—Yes/No | | √ |
| Broke rules | Number of years cleared field illegally 2008–2014 | Cont. (sqrt+1) | | √ |

## 3. Results

### 3.1. Perceptions of Access to Land

Access to land comes out as a significant concern across the 20 villages surveyed, with 62% of the 1129 respondents disagreeing with the statement that their current land access was enough to meet their household needs; and 47% of respondents stating their future access would not be enough (Figure 3).

**Figure 3.** Perceptions of respondents on current and future access to land across the full dataset, and the core payment for environmental services (PES) villages.

Results within the four core PES villages are similar to the landscape-wide results, with 60% of households feeling they did not have enough land for their current needs and 46% stating they believed their future needs would not be met (Figure 3). Furthermore, the regression results allow us to identify significant factors influencing household perceptions of land access issues across the landscape (Figure 4).

While few of the livelihood strategies outlined in the full model were supported as significant influences, household characteristics come out as the factors with the biggest magnitude of effects on perceptions. In fact, households with higher economic status feel less concerned about current and future access issues across the landscape and within the PES subset: Richer families were 2.6 times more likely to feel secure about their current access to land (*p*-value < 0.001), and 1.40 times more likely to feel similarly with regards to their future access (*p*-value < 0.001), when compared to lower income families across the 20 villages.

Families with more land felt more secure in their current access in both the 20 villages and the PES village subset, being respectively 3.10 and 2.6 times more likely to strongly perceive current access as secure. However, there is no evidence supporting the expectation that current ownership translates into perceived security of land access. Larger households feel concerned about access to land, with odd ratios indicating they are 0.5 and 0.6 times (*p*-value < 0.01 and 0.001) more likely to feel negatively about their current and future access across both the whole landscape and the PES villages.

**Figure 4.** Factors influencing perceptions of current and future access to land across the landscape (20 villages) and the core PES villages (4 villages). Results are based on the ordinal regressions of five level response variables of agreement to Likert-type item statements (see Appendix B for full results). The legend illustrates size of symbols corresponding to coefficients' *p*-values; green symbols indicate a positive correlation to response while red symbols indicate a negative correlation to response.

Selling labour was also correlated with perceived lack of current and future access to land across the 20 villages. There are significantly higher percentages of households selling labour in villages outside PAs, and also in villages affected by ELCs (Appendix C). The direction of the causality here can be questioned, as active ELCs can provide an opportunity for nearby villagers to sell labour while at the same time potentially being the source of land evictions pushing families into selling labour. This correlation might have been aggravated in villages where higher levels of education were available, attracting newcomers:

> "Many newcomers come here from other provinces because there is a lack of land in their home area. They now make 8% of the population of the commune; but they come to Bangkan because of the school and health centre. They need new rice land and timber for their house".
>
> —Rieb Roy Commune Council Chief, man, 56 years-old, Bankang village

There is no evidence that variance between each village is significant, based on testing the effect of the addition of a random effect to the models. Yet patterns in the availability of school years in a village show that the level of connectivity plays an important role in shaping household perceptions of land access. In fact, households in village with higher levels of education available were respectively 0.4 and 0.7 times more likely to feel insecure about current (*p*-value < 0.001), and future land access (*p*-value < 0.001) across the 20 villages. A similar effect is observed in the PES subset for current access, however connectivity is positively correlated to perceived future access within the PES villages. This might be explained by the continuous expansion of conservation activities in some of the four villages:

> "Since the CPA started there is less destruction. But the CPA cannot protect everything everywhere; new families need more land, and outsiders and companies still come cut trees in the PA area. Thankfully, there are plans for the CPA to expand."
>
> —Man, 48 years-old, Tmatboey village

Overall, the presence of conservation and development interventions have smaller effects on the perceptions than household characteristics. However, the presence of PAs emerged as a factor perceived to curtail people's current access to land. Households inside PAs were 0.70 times more likely than households located outside PAs to consider that current access was not enough for their household needs ($p$-value < 0.05).

> "I'm not satisfied with my current land because of the CPA and PA rules. I only have two hectares for my 4 children. But I have requested more and next year it will be better."
>
> —Man, 55 years-old, Prey Veng village

In turn, presence of ELCs is associated with perceived lack of future access to land; households in villages affected by ELCs were 0.80 more likely to be concerned about their future access to land in three years than households in villages not affected by ELCs ($p$-value < 0.01).

### 3.2. Perceptions of Participation and Trust in Land Management

Across the 20 villages, a majority of respondents had positive perceptions of their participation in land management decisions (79%) and trusted the commune council to implement land laws (53%) (Figure 5). Within the PES subset, respondents felt less involved in land management decisions than across the landscape in general (60%); however, over half the families (55%) perceived community protected area committees to be effective in their role of protecting the forest.

**" I feel involved in decision-making about land use management"**

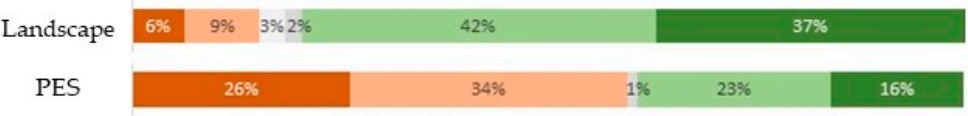

**"I feel I can trust the village committees to implement land policies"**

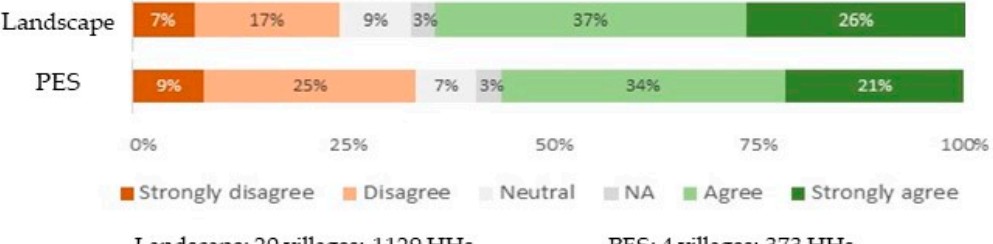

**Figure 5.** Perceptions of respondents on participation and trust in land management across the full dataset, and the core PES villages.

Fewer factors influenced the perceptions of participation and trust in land management (Figure 6). Richer families are 1.8 times more likely to feel involved in land management compared to poorer families across the 20 villages ($p$-value < 0.001). Families currently owning a larger area of land felt more involved in management decisions as well; however, the direction of causation is unclear. One may argue that those with the power to influence land management decisions effectively grant themselves high amounts of land, but the converse may also be true. Both economic status and land owned did not affect the perceptions of trust in the commune council, thus posing the question whether the laws implemented by the commune council level are effectively translated into village-level management decisions—or whether a different set of rules, and consequently rulers, are in place.

**Figure 6.** Factors influencing perceptions of participation and trust in land management across the landscape (20 villages) and the core PES villages (4 villages). Results are based on the ordinal regressions of five level response variables of agreement to Likert-type item statements, modelled to a priori predictor variables (see Appendix B for full results). The legend illustrates size of symbols corresponding to coefficients' *p*-values; green symbols indicate the positive correlation to response while red symbols indicate a negative correlation to response.

Residents of more connected villages felt much less involved in the land management decisions in their village and trusted their commune councils less. These two points resonate with comments from Bangkan village pointing to connectivity and related in-migration imposing increased pressure on land access, as well as to the idea that the commune council are less important implementers of land decisions than village authorities. These factors do not emerge within the PES subset, which suggests that the in-migration and development limits imposed on the PES villages could be effective in preventing these pressures.

Within the subset of PES villages, educated households were 1.5 times more likely to perceive positively their involvement in land management decisions in their village than less educated families (*p*-value < 0.01). This is not surprising as several roles within the CPA committee require a person to be able to read and write. Here again, the analysis points to those being in charge of land management being more educated rather than richer. In fact, wealthy families were 0.4 times less likely to trust their CPA committee than poorer ones. Last, resin-tappers are less likely to feel involved in land decisions.

## 4. Discussion

### 4.1. Power and Perception of Land Issues

Overall, household characteristics are consistently influencing the perceptions of land issues more than village level variables and interventions, with a general sense of insecurity in maintaining access to land. Similar trends are observed with regards to issues of current and future access in landscape and PES villages. In fact, economic status is the only factor perceived to assure current and future access to land.

Current land possession does not assure future access. This could be a result of the changes in land rights between the 1992 and 2011 Land Law, as customary rights of possession are not honoured any more [32]. This is problematic as even if a family currently commands enough land and resources, translating these changes into observable and accessible economic status takes time and may not be helpful in securing future land [65,66]. Economic status is often correlated with political leverage in Cambodia [67,68]. In fact, Neef et al. (2013) suggested that the existing land sector configuration is dictated by ruling elites to promote political legitimacy through the politics of patronage. Local elites,

richer and with more political agency, are thus also more likely to be involved in decisions about land management in the village and consequently to allocate themselves larger land areas.

Patterns of perceptions of land management and trust are different in PES villages compared to landscape villages. In particular, more educated households are more engaged in land management, rather than richer households. While richer households still feel secure in their future land access, they also mistrust the CPA committees in the PES villages. This suggests that the land management system put in place in the PES villages has potentially circumvented, or at least frustrated, co-option of land by economic interests locally. However, the negative perceptions of land management among resin tappers, now a minority livelihood group in PES villages, suggest that participation may not include all groups within the PES villages.

*4.2. Future Access to Land in a Changing Landscape*

Unexpectedly, households in villages affected by ELCs did not worry about their current access to land. This is possibly because not all ELC areas are fully active. However, respondents in ELC affected villages strongly perceived their future access to be threatened. This supports an abundance of research highlighting the negative impacts of ELC violations of human rights through evictions [35,69,70]. The effect of ELCs on perceived future threats can be aggravated by increased migration, as most ELC workers are migrants coming from outside the province [53]. This is especially the case for regional epicentre villages, which separately attract migrant families fleeing from environmental shocks to remote provinces where new land is available [71]. Connectivity and accessibility can make people more worried about land access and trust in the management process for allocating land to newcomers [72,73], who commune councils have to provide with land by law [31].

This trend is however not found in well-connected PES villages, where households feel more secure about future land access. This echoes the lack of negative effects of village connectivity on land management and trust issues in PES villages, compared to landscape ones. This strengthens the suggestion that PES villages may be sheltered from external land pressures, for example through the regulation of migration and land expansion. The more connected PES villages also feature a high level of conservation activities, relatively increasing trust and participation [73].

Complex dynamics underpin perceptions within PA and PES villages. Households in PAs felt less secure in their current access to land, probably because of the rules that limit resource use and land to five hectares. Unexpectedly, the presence of a PA is not significant in predicting security in future access to land. Thus, despite being limiting household land to 5 hectares, PA and conservation rules seem to have provided a system through which land demands can be processed fairly. Although larger households remain insecure as the process of getting land is too slow to meet their growing needs.

Under the fast-changing land dynamics, it is not possible to conclude whether the conservation programme in place will be able to maintain some of the land access security it currently seems to provide. Despite the CPA committees being established through an iterative community-elected process, participation does not seem to be high. Addressing the mistrust that we identified, and further inclusion of all village social groups in land management processes, should be the priorities in order to effectively maintain and increase security of land access for the communities involved.

## 5. Conclusions

While perceptions are one type of information that is often dismissed as anecdotal by those arguing for evidence-based conservation, they should be an integrated part of the information used to advise conservation policy [74]. This study shows that using a quantitative assessment of perception indicators allows the analysis of individual subjective experiences while situating these experiences within their social, economic and intervention contexts to untangle significant patterns across a larger landscape [75].

The current analysis sheds light on the overall concerns about land issues across villages in Northern Cambodia, especially in terms of perceived future access to land and trust in local land

management authorities. Household characteristics, and especially economic status, are the main factors influencing the perceptions of land issues. Yet interventions also affect perceptions; especially with regards to the negative effect of development pressures such as ELCs and population growth on perceived future access to land. While the presence of PAs does not calm insecurity about future land access, the evidence suggests that higher levels of conservation activity and land management in some PES villages can circumvent landscape drivers—at least for now. More work towards social inclusion and representation of minority groups in CPA village committees is however needed to ensure the long-term sustainability of the conservation interventions and the well-being of communities involved.

**Author Contributions:** E.B., T.C. and E.J.M.-G. designed the conceptualization, the research design for the empirical analysis. E.B. designed and performed the data collection, the content and statistical analysis, writing including review and edition. T.C. and E.J.M.-G. supervised, validated and reviewed the analysis and writing.

**Funding:** This research was funded by the UK government's Economic and Social Research Council and Department for International Development under grant ES/J018155/1, by the Wildlife Conservation Society, by the UK government's Natural Environment Research Council under grant NE/P004210/1. The views expressed in this article are not necessarily those of 3ie or its members. E.B. gratefully acknowledges funding from Imperial College London's Rector's Scholarship and the Canadian Centennial Scholarship Fund.

**Acknowledgments:** We also thank staff from the Wildlife Conservation Society for their expert contributions and the local population of the 20 villages who kindly gave time to contribute to this study.

**Conflicts of Interest:** The authors declare no conflict of interest. The funders had no role in the design of the study; in the collection, analyses, or interpretation of data; in the writing of the manuscript, or in the decision to publish the results.

## Appendix A. Village Information

**Table A1.** Statistics of villages surveyed; four core PES villages in bold.

| Villages | Interviews | PA Status | ELC Affected | Distance to ELC (km) | Population (HHs) | Time to Provincial Capital (h) | Years of Schooling |
|---|---|---|---|---|---|---|---|
| Bangkan | 49 | 0 | 1 | 0 | 199 | 7 | 9 |
| Kunapheap | 43 | 1 | 1 | 0 | 143 | 4 | 6 |
| Sambour | 53 | 1 | 1 | 0 | 121 | 3 | 6 |
| Slaeng Toul | 45 | 0 | 1 | 0 | 64 | 3.5 | 6 |
| Srae | 46 | 0 | 1 | 0 | 102 | 5.5 | 6 |
| Kdak | 47 | 0 | 1 | 1.02 | 411 | 2.5 | 6 |
| Svai Damnak Chas | 42 | 0 | 1 | 1.5 | 187 | 2 | 6 |
| Rumchek | 45 | 0 | 0 | 1.57 | 184 | 2.5 | 6 |
| Antil | 57 | 1 | 1 | 2.42 | 215 | 6 | 6 |
| Kralas Peas | 50 | 1 | 1 | 2.70 | 295 | 2 | 6 |
| **Prey Veng** | 62 | 1 | 1 | 2.92 | 85 | 4 | 3 |
| Reaksmei | 57 | 1 | 0 | 3.15 | 137 | 2 | 6 |
| Chomsrae | 55 | 1 | 0 | 3.74 | 216 | 5 | 6 |
| **Dongplat** | 110 | 1 | 0 | 5.10 | 228 | 1.5 | 9 |
| Mrech | 41 | 0 | 1 | 5.90 | 125 | 3 | 3 |
| **Tmatboey** | 139 | 1 | 0 | 6.46 | 286 | 1.5 | 9 |
| **Narong** | 64 | 1 | 0 | 7.66 | 150 | 2 | 6 |
| Srea Veal | 42 | 0 | 0 | 9.24 | 170 | 3 | 6 |
| Phneak Roluek | 42 | 0 | 0 | 10.43 | 131 | 5 | 6 |
| Robohn | 41 | 1 | 0 | 29.49 | 83 | 6 | 6 |

**Table A2.** Number of villages per treatment.

| | No ELC | ELC |
|---|---|---|
| Non PA | 3 | 6 |
| PA | 6 | 5 |

**Table A3.** Number of households interviewed per treatment.

|        | No ELC | ELC |
|--------|--------|-----|
| Non PA | 131    | 270 |
| PA     | 466    | 262 |

## Appendix B. Ordinal Regression Results

Log-odds coefficients and their exponentiated format into odds ratios are presented, along with cut-points of the threshold values between the adjacent levels of the response variable. An odd ratio equal to 1 indicates a neutral effect of a given variable on the perception of mixed forests. Odds ratio > 1 represents a positive effect of the explanatory variable on the response level, while an odd ratio < 1 represents an inverse relationship between the explanatory variable and the dependent variable levels. Threshold (or cut-point) values between the five categories are given for each regression.

*Appendix B.1. Results of Ordinal Regressions for Dataset of 20 Villages*

**Table A4.** Ordinal regression with a logit link function of perception of current access to land to predictor variables for the 20 surveyed villages. Significance values: "*" = $P < 0.05$; "**" = $P < 0.01$; "***" = $P < 0.001$.

| Variables              | Estimate | Std. Error | *p* | Odds Ratio | CI (2.5%) | CI (97.5%) |
|------------------------|----------|------------|-----|------------|-----------|------------|
| PA presence            | −0.30    | 0.12       | *   | 0.70       | 0.60      | 0.90       |
| HH size                | −0.63    | 0.16       | *** | 0.50       | 0.40      | 0.70       |
| Economic status        | 0.96     | 0.14       | *** | 2.60       | 2.00      | 3.50       |
| Land owned             | 1.13     | 0.18       | *** | 3.10       | 2.20      | 4.40       |
| Sell labour            | −0.33    | 0.12       | **  | 0.70       | 0.60      | 0.90       |
| School years in village| −0.90    | 0.18       | *** | 0.40       | 0.30      | 0.60       |
| 1\|2                   | −0.14    | 0.67       |     |            |           |            |
| 2\|3                   | 1.50     | 0.67       |     |            |           |            |
| 3\|4                   | 1.50     | 0.67       |     |            |           |            |
| 4\|5                   | 2.93     | 0.68       |     |            |           |            |

**Table A5.** Ordinal regression with a loglog link function of perception of future security of access to land to predictor variables for the 20 surveyed villages. Significance values: "*" = $P < 0.05$; "**" = $P < 0.01$; "***" = $P < 0.001$.

| Variables              | Estimate | Std. Error | *p* | Odds Ratio | CI (2.5%) | CI (97.5%) |
|------------------------|----------|------------|-----|------------|-----------|------------|
| ELC presence           | −0.22    | 0.08       | **  | 0.80       | 0.70      | 0.90       |
| HH size                | −0.55    | 0.11       | *** | 0.60       | 0.50      | 0.70       |
| Female headed HH       | −0.20    | 0.12       |     | 0.80       | 0.60      | 1.00       |
| Economic status        | 0.37     | 0.08       | *** | 1.40       | 1.20      | 1.70       |
| Sell labour            | −0.17    | 0.07       | *   | 0.80       | 0.70      | 1.00       |
| School years in village| −0.50    | 0.13       | *** | 0.60       | 0.50      | 0.80       |
| 1\|2                   | −2.14    | 0.47       |     |            |           |            |
| 2\|3                   | −1.25    | 0.47       |     |            |           |            |
| 3\|4                   | −0.90    | 0.47       |     |            |           |            |
| 4\|5                   | 0.56     | 0.47       |     |            |           |            |

**Table A6.** Ordinal regression with a cauchit link function of perception of participation in land management decisions to predictor variables for the 20 surveyed villages. Significance values: "***" = *P* < 0.001.

| Variables | Estimate | Std. Error | *p* | Odds Ratio | CI (2.5%) | CI (97.5%) |
|---|---|---|---|---|---|---|
| HHH education | 0.30 | 0.08 | *** | 1.40 | 1.20 | 1.60 |
| Economic status | 0.59 | 0.12 | *** | 1.80 | 1.40 | 2.30 |
| School years in village | −0.62 | 0.18 | *** | 0.50 | 0.40 | 0.80 |
| 1\|2 | −3.94 | 0.78 | | | | |
| 2\|3 | −0.97 | 0.57 | | | | |
| 3\|4 | −0.60 | 0.56 | | | | |
| 4\|5 | 1.41 | 0.57 | | | | |

**Table A7.** Ordinal regression with a log-log link function of perception of trust in commune council to predictor variables for the 20 surveyed villages. Significance values: "*" = *P* < 0.05; "***" = *P* < 0.001.

| Variables | Estimate | Std. Error | *p* | Odds Ratio | CI (2.5%) | CI (97.5%) |
|---|---|---|---|---|---|---|
| Cash crop farmer | 0.37 | 0.15 | * | 1.50 | 1.10 | 2.00 |
| School years in village | −0.75 | 0.17 | *** | 0.50 | 0.30 | 0.70 |
| 1\|2 | −4.53 | 0.49 | | | | |
| 2\|3 | −3.08 | 0.48 | | | | |
| 3\|4 | −2.66 | 0.48 | | | | |
| 4\|5 | −0.99 | 0.47 | | | | |

**Table A8.** Ordinal regression with a c-log-log link function of perception of trust in police for security predictor variables for the 20 surveyed villages. Significance values: "***" = *P* < 0.001.

| Variables | Estimate | Std. Error | *p* | Odds Ratio | CI (2.5%) | CI (97.5%) |
|---|---|---|---|---|---|---|
| PA presence | −0.29 | 0.08 | *** | 0.70 | 0.60 | 0.90 |
| School years in village | 0.39 | 0.12 | *** | 1.50 | 1.20 | 1.90 |
| 1\|2 | −1.22 | 0.32 | | | | |
| 2\|3 | 0.03 | 0.31 | | | | |
| 3\|4 | 0.20 | 0.31 | | | | |
| 4\|5 | 1.26 | 0.31 | | | | |

*Appendix B.2. Results of Ordinal Regressions for Subset of Four Core PES Villages*

**Table A9.** Ordinal regression a logit link function of perception of current access to land to predictor variables within four core PES villages. Significance values: "**" = *P* < 0.01; "***" = *P* < 0.001.

| Variables | Estimate | Std. Error | *p* | Odds Ratio | CI (2.5%) | CI (97.5%) |
|---|---|---|---|---|---|---|
| Illegal clearings (yrs) | 0.39 | 0.12 | ** | 1.5 | 1.2 | 1.9 |
| HH size | −0.78 | 0.28 | ** | 0.5 | 0.3 | 0.8 |
| Economic status | 1.23 | 0.29 | *** | 3.4 | 1.9 | 6.0 |
| Land owned | 0.97 | 0.34 | ** | 2.6 | 1.4 | 5.1 |
| Sell labour | −0.60 | 0.21 | ** | 0.5 | 0.4 | 0.8 |
| School years in village | −1.25 | 0.25 | *** | 0.3 | 0.2 | 0.5 |
| 1\|2 | −0.40 | 1.17 | | | | |
| 2\|4 | 1.28 | 1.17 | | | | |
| 4\|5 | 2.69 | 1.18 | | | | |

**Table A10.** Ordinal regression a probit link function of perception of future security of access to land to predictor variables within four core PES villages. Significance values: "**" = P < 0.01; "***" = P < 0.001.

| Variables | Estimate | Std. Error | p | Odds Ratio | CI (2.5%) | CI (97.5%) |
|---|---|---|---|---|---|---|
| HH size | −0.64 | 0.15 | *** | 0.5 | 0.4 | 0.7 |
| Economic status | 0.67 | 0.14 | *** | 1.9 | 1.5 | 2.6 |
| School years in village | −0.37 | 0.14 | ** | 0.7 | 0.5 | 0.9 |
| 1|2 | −1.09 | 0.62 | | | | |
| 2|3 | −0.20 | 0.62 | | | | |
| 3|4 | 0.11 | 0.62 | | | | |
| 4|5 | 1.03 | 0.62 | | | | |

**Table A11.** Ordinal regression a cauchit link function of perception of participation in land management decisions to predictor variables within four core PES villages. Significance values: "**" = P < 0.01.

| Variables | Estimate | Std. Error | p | Odds Ratio | CI (2.5%) | CI (97.5%) |
|---|---|---|---|---|---|---|
| Resin tapper | −0.53 | 0.20 | ** | 0.6 | 0.4 | 0.9 |
| HHH education | 0.39 | 0.14 | ** | 1.5 | 1.1 | 1.9 |
| 1|2 | −6.33 | 1.62 | | | | |
| 2|3 | −1.49 | 0.37 | | | | |
| 3|4 | −0.98 | 0.32 | | | | |
| 4|5 | 0.93 | 0.30 | | | | |

**Table A12.** Ordinal regression a logit link function of perception of trust in community protected area (CPA) committees to predictor variables within four core PES villages. Significance values: "**" = P < 0.01; "***" = P < 0.001.

| Variables | Estimate | Std. Error | p | Odds Ratio | CI (2.5%) | CI (97.5%) |
|---|---|---|---|---|---|---|
| HH size | −0.77 | 0.26 | ** | 0.5 | 0.3 | 0.8 |
| Economic status | −0.85 | 0.26 | *** | 0.4 | 0.3 | 0.7 |
| Land owned | 1.44 | 0.33 | *** | 4.2 | 2.2 | 8.1 |
| 1|2 | −5.16 | 1.01 | | | | |
| 2|3 | −3.35 | 0.98 | | | | |
| 3|4 | −3.01 | 0.98 | | | | |
| 4|5 | −1.34 | 0.97 | | | | |

## Appendix C. Household Characteristics and Livelihood Strategies Across Treatment Types

**Table A13.** Household characteristics and livelihoods strategies for households inside compared to outside PAs. *T*-tests of difference applied to values for households between treatments. Significance values: "ns" = non-significant; "." = P < 0.1; "*" = P < 0.05; "**" = P < 0.01; "***" = P < 0.001.

| Household Characteristics | Non-PA | PA | p |
|---|---|---|---|
| Households | 401 | 728 | |
| Household size | 5.6 | 5.7 | ns |
| Household head education (yrs) | 2.2 | 2.6 | * |
| Household head age (yrs) | 41.3 | 44.7 | *** |
| Female-headed households (%) | 10% | 11% | ns |
| *Household productivity* | | | |
| Socio-economic status score | 11.8 | 12.7 | *** |
| Rice harvest (kg) | 2574 | 3315 | *** |
| Food security (kg) | 1179 | 1530 | ** |
| Total rice land area (ha) | 1.83 | 1.99 | ns |
| Number of Cattle (heads) | 2.08 | 2.26 | ns |

**Table A13.** *Cont.*

| Household Characteristics | Non-PA | PA | *p* |
|---|---|---|---|
| *Livelihood strategies* | | | |
| Resin-tapper (%) | 21% | 50% | *** |
| Rice farmer (%) | 100% | 100% | ns |
| Rice shifting cultivation (%) | 19% | 15% | *** |
| Cash crop (%) | 23% | 13% | *** |
| Employed (%) | 9% | 10% | ns |
| Employed admin gvt (%) | 4% | 4% | ns |
| Employed NGO (%) | 0% | 4% | ** |
| Village service or shop (%) | 29% | 28% | ns |
| Sell labour (%) | 64% | 47% | *** |

**Table A14.** Household characteristics and livelihoods strategies for households affected by ELCs compared to households not affected. *T*-tests of difference applied to values for households between treatments. Significance values: "ns" = non-significant; "." = $P < 0.1$; "*" = $P < 0.05$; "**" = $P < 0.01$; "***" = $P < 0.001$.

| Household Characteristics | Non-ELC | ELC | *p* |
|---|---|---|---|
| Households | 597 | 532 | |
| Household size | 5.7 | 5.6 | ns |
| Household head education (yrs) | 2.4 | 2.5 | ns |
| Household head age (yrs) | 44.6 | 42.4 | ns |
| Female-headed households (%) | 11.6% | 9.4% | ns |
| *Household productivity* | | | |
| Poverty | 12.9 | 11.8 | * |
| Rice harvest (kg) | 3502 | 2547 | *** |
| Food security (kg) | 1589 | 1199 | * |
| Total rice land area (ha) | 1.9 | 1.9 | ns |
| Number of Cattle (heads) | 2.3 | 2.1 | ns |
| *Livelihood strategies* | | | |
| Resin-tapper (%) | 53% | 25% | *** |
| Rice farmer (%) | 100% | 100% | ns |
| Rice shifting cultivation (%) | 14% | 19% | * |
| Have > 1 hectare of paddyfields (%) | 85% | 80% | * |
| Cash crop (%) | 20% | 12% | * |
| Employed (%) | 10% | 10% | ns |
| Employed admin gvt (%) | 5% | 3% | ns |
| Employed NGO (%) | 3% | 2% | * |
| Village service or shop (%) | 29% | 28% | ns |
| Sell labour (%) | 43% | 64% | *** |
| Mini-tractor (%) | 73% | 61% | *** |

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
