# Peer review of "Investigating Perceptions of Land Issues in a Threatened Landscape in Northern Cambodia"

_sustainability, doi:10.3390/su11215881_

Round 1

Reviewer 1 Report

The paper investigates household perceptions of land issues in relation to conservation (Protected Areas) and development interventions (Economic Land Concessions) in Cambodia. The authors conclude that household characteristics rather than village contexts are the main factors influencing perceptions of land issues and that there is no substantial effect of the PA's or ELCs on calming insecurities related to land, but that in some villages, PES programs may alleviate this concern. The study is sound and well designed and the analysis is robust. I do however feel that the interpretation of the results could be revised - the more prominent results could be be better highlighted both in the main text and in the abstract. I also feel like not all interpretations are well supported by the data and overall I recommend a through revision of the discussion section.

Not being an expert in human perceptions of land issues I did find that one interesting finding that could be granted more attention in the manuscript is the discrepancy in participation in management between PES and Landscape viallges. I know this was not the core aim of the analysis, but could it say something about the way that PES was implemented (maybe rather top-down)? My expectation would have been that in PES villages, people feel more involved in the management through their activities according to the PES scheme. I feel like this could yeild more and interesting material for discussion. 

I wonder if the age of some the PAs and ELC may mean that more engrained perceptions of villagers have not yet changed after the establishment? How long might it take after a PA has been establish for the people to change their view about it?

I think the text could be shortened overall a bit, to the benefit of highlighting the more interesting fidings. Some small suggestions below.

Smaller comments:

Figure 1  is not low quality, text is hard to read and the colors very hard to distinguish. Are there two PAs in the study region as described in the text, or three, as suggested by the figure? Also, are there ECL in PAs? these are hardly visible in the figure.

The paper uses  A LOT of acronyms, and for this reason the text is really hard to follow. Some of the acronyms are not even spelled out the first time they are used, and the reader ends up having to go back and forth multiple times to follow the text. I suggest using acronyms for only a couple of the most often used expressions such as the PA, PES or ECL but spell out the rest (such as the non-timber forest products, the administrative bodies etc). Maybe as a rule of thumb, if something appears less than 7-8 times in a manuscript then spell it out.....?

Lines 188-198 could be shortened, or moved to the Appendix. Likewise some of the villagers quotes could be reduced. 

Line 218 onwards: assigning the ELC seems a bit arbitrary, but the concern that not all are active seems valid. Could the analysis be performed including just the geographical delineation of ELC and then including villagers perceptions as a robustness check. Also, what does fig 1 depicts the ELC just geographically or as described here?

Line 252-253 the references are not formatted like the rest, and assume they miss from the ref list.

Line 387 Typo regarding resin tappers: more OR less

Line 424 I am not sure I follow which of the results best support this statement.

Congratulations on your work!

Author Response

COMMENTS FROM REFEREE 1

Comments and Suggestions for Authors

The paper investigates household perceptions of land issues in relation to conservation (Protected Areas) and development interventions (Economic Land Concessions) in Cambodia. The authors conclude that household characteristics rather than village contexts are the main factors influencing perceptions of land issues and that there is no substantial effect of the PA's or ELCs on calming insecurities related to land, but that in some villages, PES programs may alleviate this concern.

The study is sound and well designed and the analysis is robust. I do however feel that the interpretation of the results could be revised - the more prominent results could be better highlighted both in the main text and in the abstract. I also feel like not all interpretations are well supported by the data and overall I recommend a thorough revision of the discussion section.

Response: We have restructured and revised the discussion to highlight two main findings: the role of household-level economic power in shaping perceptions of land issues, and the need for fairer land management process to support current PES benefits. For example, we are more straightforward in mentioning:

Under the fast-changing land dynamics, it is not possible to conclude whether the conservation programme in place will be able to maintain some of the land access security it currently seems to provide. Despite the CPA committees being established through an iterative community-elected process, participation does not seem to be high. Addressing the mistrust we identified, and further inclusion of all village social groups in land management processes, should be priorities in order to effectively maintain and increase security of land access for the communities involved.” P.7-8

Please refer to our response below with regards to perception in management in PES villages.

Not being an expert in human perceptions of land issues I did find that one interesting finding that could be granted more attention in the manuscript is the discrepancy in participation in management between PES and Landscape villages. I know this was not the core aim of the analysis, but could it say something about the way that PES was implemented (maybe rather top-down)? My expectation would have been that in PES villages, people feel more involved in the management through their activities according to the PES scheme. I feel like this could yield more and interesting material for discussion. 

Response: We agree with this observation and have revised the discussion section about the perceptions in PES villages. As well as the new text quoted above, we have also added the following:

“Patterns of perceptions of land management and trust are different in PES villages compared to landscape villages. In particular, more educated households feel more engagement in land management, rather than richer households. While richer households still feel secure in their future land access, they also mistrust the CPA committees in the PES villages. This suggests that the land management system put in place in the PES villages has potentially circumvented, or at least frustrated, co-option of land by economic interests locally. However, the negative perceptions of land management among resin tappers, now a minority livelihood group in PES villages, suggest that participation may not include all groups within the PES villages.” P.7

I wonder if the age of some the PAs and ELC may mean that more engrained perceptions of villagers have not yet changed after the establishment? How long might it take after a PA has been establish for the people to change their view about it?

Response: This is an interesting point. KPWF was gazetted in 1993, and PVPF in 2002, although both PAs had a low level of PA management (implementation of compliance rules) until the involvement of WCS in 2005. Thus the age of both PAs would be practically the same in effective terms, although management effort varies according to villages, and with the presence of PES.

With regards to ELCs, our research has shown that the period of time after the start of activities in an ELC seems to affect perceptions of land issues, rather than the decree (or in other words legal age) of the ELC. For this reason, we have corrected our model variable for the presence of ELC to reflect being within an active ELC, rather than being within the geographical boundary of an ELC regardless of activity.

It is logical that age of an intervention may correlate to change in engrained perceptions of its efficiency, although this falls beyond the scope of this research and data. We nonetheless note that PA or ELC management and activities are not linear, hence perceptions are likely to change at different points in time.

I think the text could be shortened overall a bit, to the benefit of highlighting the more interesting findings. Some small suggestions below.

Response: We have shortened the text overall, including in introduction, methods, results and discussion.

Smaller comments:

Figure 1: is not low quality, text is hard to read and the colors very hard to distinguish. Are there two PAs in the study region as described in the text, or three, as suggested by the figure? Also, are there ECL in PAs? these are hardly visible in the figure.

Response: Thanks for pointing this out. We have revised Figure 1 to be more easily read, by resizing the main map and accentuating the ELC boundaries.

There are three PAs in Preah Vihear, however only two are considered in this study as the third is unmanaged. We have specified this in the text and explained that the Boeng Per Wildlife Sanctuary is considered mainly as a “paper park”.

In line with consideration of defining presence/absence of intervention variables according to presence of activities, the village of Svay Demnak Chas is not considered as a PA village, but as a non-PA village (see Supp Mat 1 for village information).

“A third PA is present in the south of the landscape, however is unmanaged and considered largely as a paper park [43]The he Boeng Per Wildlife Sanctuary (BPWS) is by law under the management of the MoE, yet has seen insufficient law enforcement activities and was identified as the protected area second most threatened by land encroachment in Cambodia [44].” P. 4

The paper uses A LOT of acronyms, and for this reason the text is really hard to follow. Some of the acronyms are not even spelled out the first time they are used, and the reader ends up having to go back and forth multiple times to follow the text. I suggest using acronyms for only a couple of the most often used expressions such as the PA, PES or ECL but spell out the rest (such as the non-timber forest products, the administrative bodies etc). Maybe as a rule of thumb, if something appears less than 7-8 times in a manuscript then spell it out.....?

Response: Thanks for raising this, we have spelt out most acronyms and only kept those being repeated several times. We have kept acronyms for the main terms: PAs, PES, ELCs, and CPA.

Lines 188-198 could be shortened, or moved to the Appendix. Likewise some of the villagers quotes could be reduced. 

Response: We have shortened this paragraph.

Line 218 onwards: assigning the ELC seems a bit arbitrary, but the concern that not all are active seems valid. Could the analysis be performed including just the geographical delineation of ELC and then including villagers perceptions as a robustness check. Also, what does fig 1 depicts the ELC just geographically or as described here?

Response: Analysing the geographical presence of ELCs was originally our intention, until we realized results were misaligned with other triangulated evidence from quantitative surveys and qualitative research. This is because several ELCs in Cambodia are granted on an arbitrary basis to owners which proceed to logging some of the area, but not the entire concession (see Beauchamp et al 2017, Milne et al, 2015). We thus corrected the variable to reflect the proximity to the activities of an ELC rather than its geographical presence. This assignation was based on over 2 years of research in the study villages, hence we believe this represents an evidence-based decision.

Figure 1 depicts ELCs' geographical boundaries, as obtaining spatial information on their operations would be very difficult. We have specified this point in Figure 1’s legend.

“Figure 1. Locations of the surveyed villages, boundaries of Economics Land Concessions (ELCs) and Protected Areas (PAs) in the province of Preah Vihear.” P. 3 

We also mention in the text:

While PA presence can be straightforwardly determined by the village location inside or outside the PA, not all declared ELCs are active, or even active across their entire area [60]. Presence of ELCs was thus determined as whether an active ELC was located within 3 km of the village, with confirmation from field surveys and reports from villagers. In this sense, ELC presence was determined by geographical presence along with villagers’ perceptions of ELC presence and based on previous research (see Supp. Mat 1) [56], [61].” P. 6

Line 252-253 the references are not formatted like the rest, and assume they miss from the ref list.

Line 387 Typo regarding resin tappers: more OR less

Response: Thanks, we have reformatted and corrected these two points.

Line 424 I am not sure I follow which of the results best support this statement.

Response: We agree, and we have revised the section to reflect the duality of perceptions about PES and PAs. This is in line with other prior comments about revising the discussion. We have revised the sub-heading to “Future access to land in a changing landscape”.  The revised discussion develops this idea, for example:

“Complex dynamics underpin perceptions within PA and PES villages. Households in PAs felt less secure in their current access to land, probably due to rules that limit resource use and land to five hectares. Unexpectedly, the presence of a PA is not significant in predicting security in future access to land. Thus despite being limiting household land to 5 hectares, PA and conservation rules seem to have provided a system through which land demands can be processed fairly. Although larger households remain insecure as the process of getting land is too slow to meet their growing needs.” P. 7

Congratulations on your work!

Response: Thank you.

Reviewer 2 Report

Reviewed paper focuses on the perceptions of the local communities on the land issues connected with economic land concession and protected areas in Cambodia. Authors assumed perception indicators of land issues and studied the opinion statements of selected respondents. They focused on perceived current and future access to land, participation in land management decisions and trust in local authorities implementation of the land rules. Authors surveyed 1129 households across 20 villages in the province of Praeh Vihear in 2014. Data was tested with correlation matrix and VIF test. Authors used ordered logistic mixed regression models in order to analyse the Likert scale indicators and took other steps and procedures to generate a final model.

The paper concerns significant and insufficiently developed in literature issue – the real effects of the governmental and local regulations concerning land use and land management in order to protect the environment.  Protected areas are being set all over the world and bring many effects, having impact on the quality of life of the local people. Environment protection regulations are important and controversial. Therefore, selected topic and empirical analysis can be considered as quite important. The quality of presentation is good (paper is divided into clear sections). The main strength of the paper is the topic itself and the opinions of the local population and underlining the informal relations. Moreover, the discussion section is well developed.

There are some suggestions how the paper cold be improved:

The title is too general – it does not indicate the spatial scope of the study – Cambodia. As the paper does not develop other regions of the world, it should be added “evidence from Cambodia”. As many potential international readers are not probably familiar with the current situation in Cambodia, Authors should add a section about Cambodian economic situation, legal regulations concerning land ownership, land use, land market turnover, etc. It would make a good background and help the reader to understand the whole problem. Data was collected in 2014. It would be worth to mention if there were any changes in law or economy since then? Figure 2 is not mentioned in the text. In line 252 and 253 there are names instead of numbers in brackets. I suggest changing the titles in the Figure 3 and 5. They are not clear. Authors could develop the conclusions in order to conclude all the results.

These minor corrections will improve the quality of the paper.

Generally, the paper is clear and well organised. The references cited are correct and adequate to reflect other work. Paper also is coherent with aims and scope of the journal Sustainability.  

Author Response

COMMENTS FROM REFEREE 2

Reviewed paper focuses on the perceptions of the local communities on the land issues connected with economic land concession and protected areas in Cambodia. Authors assumed perception indicators of land issues and studied the opinion statements of selected respondents. They focused on perceived current and future access to land, participation in land management decisions and trust in local authorities implementation of the land rules. Authors surveyed 1129 households across 20 villages in the province of Praeh Vihear in 2014. Data was tested with correlation matrix and VIF test. Authors used ordered logistic mixed regression models in order to analyse the Likert scale indicators and took other steps and procedures to generate a final model.

The paper concerns significant and insufficiently developed in literature issue – the real effects of the governmental and local regulations concerning land use and land management in order to protect the environment.  Protected areas are being set all over the world and bring many effects, having impact on the quality of life of the local people. Environment protection regulations are important and controversial. Therefore, selected topic and empirical analysis can be considered as quite important. The quality of presentation is good (paper is divided into clear sections). The main strength of the paper is the topic itself and the opinions of the local population and underlining the informal relations. Moreover, the discussion section is well developed.

Response: Many thanks for your positive comments.

There are some suggestions how the paper cold be improved:

The title is too general – it does not indicate the spatial scope of the study – Cambodia. As the paper does not develop other regions of the world, it should be added “evidence from Cambodia”.

Response: We have revised the title to: “Investigating perceptions of land issues in a threatened landscape in Northern Cambodia”

As many potential international readers are not probably familiar with the current situation in Cambodia, Authors should add a section about Cambodian economic situation, legal regulations concerning land ownership, land use, land market turnover, etc. It would make a good background and help the reader to understand the whole problem. Data was collected in 2014. It would be worth to mention if there were any changes in law or economy since then?

Response: We added details about the Cambodian context especially with regards to the Land Law and land dynamics, while being mindful of the paper’s length:

“Cambodia has seen rapid development and globalization over the past decade [26], [27] , paralleled by a similarly high rate of resource depletion in terms of deforestation rates and illegal logging [28]–[30]. The 2001 Land Law initiated a process of codifying land claims by local people, indigenous communities and the private sector, marking a move towards a formal land registration system and official land titles [31]. Thus while customary land rights, or “possession rights”, were recognized under the 1992 Land Law, they were replaced in the 2001 Land Law by modern landownership where titles are required [32]. This introduction could be beneficial in supporting tenure security for local communities [33], yet its weak enforcement and co-option by the elite opened the door for the monetization of land titles. It also enabled the widespread granting of Economic Land Concessions (ELCs) as large areas of state public land were reclassified as state private, consequently weakening informal possession rights in poor rural communities [34].” P. 2-3  

We have also highlighted a change in the names of the Protected Areas: “The Preah Vihear Protected Forest was declared in 2002, and in 2018, after data collection, was split under two PAs: the Preah Roka Wildlife Sanctuary and Chhep Wildlife Sanctuary.” P. 4

Figure 2 is not mentioned in the text. In line 252 and 253 there are names instead of numbers in brackets. I suggest changing the titles in the Figure 3 and 5. They are not clear.

Response: We have reformatted the references appropriately. Figure 2 is mentioned in the text in (current) line 193.

We have revised the titles of Figures 3 and 5 as the perception statement, e.g.; “I feel I have enough land for my needs” to fit with the responses. 

Authors could develop the conclusions in order to conclude all the results.

Response: We have revised the discussion section and the conclusions, to discussion to highlight two main findings: the role of household-level economic power in shaping perceptions of land issues, and the need for fairer land management process to support current PES benefits. For example, we are more straightforward in concluding:

Under the fast-changing land dynamics, it is not possible to conclude whether the conservation programme in place will be able to maintain some of the land access security it currently seems to provide. Despite the CPA committees being established through an iterative community-elected process, participation does not seem to be high. Addressing the mistrust we identified, and further inclusion of all village social groups in land management processes, should be priorities in order to effectively maintain and increase security of land access for the communities involved.” P.7-8

We also reinforce this point in our conclusion:

“While the presence of PAs does not calm insecurity about future land access, the evidence suggest that higher levels of conservation activity and land management in some PES villages can circumvent landscape drivers – at least for now. More work towards social inclusion and representation of minority groups in CPA village committees is however needed to ensure the long-term sustainability of the conservation interventions and the well-being of communities involved.” P. 8  

These minor corrections will improve the quality of the paper.

Generally, the paper is clear and well organised. The references cited are correct and adequate to reflect other work. Paper also is coherent with aims and scope of the journal Sustainability.

Response: Thank you.
